# Polyglucose nanoparticles with renal elimination and macrophage avidity facilitate PET imaging in ischaemic heart disease

Edmund J. Keliher[1,*], Yu-Xiang Ye[1,*], Gregory R. Wojtkiewicz[1], Aaron D. Aguirre[1], Benoit Tricot[1], Max L. Senders[2,3], Hannah Groenen[2], Francois Fay[2], Carlos Perez-Medina[2], Claudia Calcagno[2], Giuseppe Carlucci[4], Thomas Reiner[4,5], Yuan Sun[1], Gabriel Courties[1], Yoshiko Iwamoto[1], Hye-Yeong Kim[1], Cuihua Wang[1], John W. Chen[1], Filip K. Swirski[1], Hsiao-Ying Wey[6], Jacob Hooker[6], Zahi A. Fayad[2], Willem J.M. Mulder[2,3], Ralph Weissleder[1,7] & Matthias Nahrendorf[1,8]

Tissue macrophage numbers vary during health versus disease. Abundant inflammatory macrophages destruct tissues, leading to atherosclerosis, myocardial infarction and heart failure. Emerging therapeutic options create interest in monitoring macrophages in patients. Here we describe positron emission tomography (PET) imaging with $^{18}$F-Macroflor, a modified polyglucose nanoparticle with high avidity for macrophages. Due to its small size, Macroflor is excreted renally, a prerequisite for imaging with the isotope flourine-18. The particle's short blood half-life, measured in three species, including a primate, enables macrophage imaging in inflamed cardiovascular tissues. Macroflor enriches in cardiac and plaque macrophages, thereby increasing PET signal in murine infarcts and both mouse and rabbit atherosclerotic plaques. In PET/magnetic resonance imaging (MRI) experiments, Macroflor PET imaging detects changes in macrophage population size while molecular MRI reports on increasing or resolving inflammation. These data suggest that Macroflor PET/MRI could be a clinical tool to non-invasively monitor macrophage biology.

[1] Center for Systems Biology and Department of Imaging, Massachusetts General Hospital and Harvard Medical School, Simches Research Building, 185 Cambridge Street, Boston, Massachusetts 02114, USA. [2] Translational and Molecular Imaging Institute and Department of Radiology, Icahn School of Medicine at Mount Sinai, One Gustave L. Levy Place, New York, New York 10029, USA. [3] Department of Medical Biochemistry, Subdivision of Experimental Vascular Biology, Academic Medical Center, Meibergdreef 9, 1105 AZ Amsterdam, The Netherlands. [4] Department of Radiology, Memorial Sloan-Kettering Cancer Center, 1275 York Avenue, New York, New York 10065, USA. [5] Department of Radiology, Weill Cornell Medical College, New York, New York 10065, USA. [6] Athinoula A. Martinos Center for Biomedical Imaging, Department of Imaging, Massachusetts General Hospital, Harvard Medical School, 149 Thirteenth Street, Charlestown, Massachusetts 02129, USA. [7] Department of Systems Biology, Harvard Medical School, 200 Longwood Avenue, Alpert 536, Boston, Massachusetts 02115, USA. [8] Cardiovascular Research Center, Massachusetts General Hospital and Harvard Medical School, Simches Research Building, 185 Cambridge Street, Boston, Massachusetts 02114, USA. * These authors contributed equally to this work. Correspondence and requests for materials should be addressed to R.W. (email: rweissleder@mgh.harvard.edu) or to M.N. (email: mnahrendorf@mgh.harvard.edu).

schaemic heart disease leads worldwide mortality statistics[1]. While conventional risk factors such as hyperlipidemia and hypertension are well understood and effectively treated, the atherosclerosis disease burden keeps growing. Macrophage function is increasingly seen as therapeutically interesting, as per genome-wide association studies showing inflammation-related mutations in patients with myocardial infarction (MI)[2] and fundamental immunology discoveries. Data obtained from animals with cardiovascular disease suggest that surplus inflammatory macrophages in either the arterial wall or ischaemic heart muscle promote tissue destruction, morbidity and death[3]. In patients, most available evidence focuses on blood monocytes, which are macrophage precursors that are easy to assay with existing technology. However, because tissue macrophage numbers and phenotypes do not necessarily correlate with blood monocytes, we need to directly survey vascular and cardiac tissues. In preclinical research, intravital microscopy and flow cytometry enable tissue macrophage quantification[4]. Yet it is not practical to biopsy atherosclerotic blood vessels and infarcted or failing hearts in patients. Quantitative macrophage imaging, currently unavailable for clinical applications, would overcome this barrier, provide better understanding of macrophages' roles in human disease, identify patients at risk for complications and evaluate emerging macrophage-targeting therapeutics.

Nanoparticles, which can be efficiently internalized by phagocytic myeloid cells[5], are a promising strategy for quantitatively and specifically imaging macrophages in human cardiovascular organs. Previously developed nanoparticles with effective macrophage uptake also have long circulation times with high blood pool activity that limit target-to-background ratios in the vascular system. Further, the long circulation times precluded the use of the clinically facile [18]F PET isotope for nanoparticle tracking, because the radioisotope decays faster ($T_{1/2}$ 110 min) than nanoparticles exit from the blood pool adjacent to the imaging target, that is, diseased cardiovascular tissues.

To solve these problems, we shrank nanoparticles to a size below the renal excretion threshold[6] and optimized biological behaviour through biocompatible chemistries. Here we describe a class of modified polyglucose nanoparticles that we named macrins. More specifically, macrins are a class of lysine-crosslinked low molecular weight carboxymethyl polyglucose polymers, each containing 22 glucose units. Macrins can be considered glycogen biomimetics that lack the central glycogenic core. For our imaging applications, we labelled a 5 nm macrin with either [18]F to create a positron emission tomography (PET) imaging agent (termed Macroflor) or a fluorochrome for correlative studies. In three animal species, we confirm rapid renal excretion of Macroflor and also show its high affinity for macrophages residing in cardiovascular organs. In a number of imaging experiments, this approach provides quantitative and specific PET data on inflammation in atherosclerotic plaque and ischaemic myocardium.

## Results
**Small nanoparticles are excreted renally.** Macroflor synthesis relied on the commercial building blocks carboxymethylated polyglucose (with an average chain length of 22) and L-lysine crosslinked through amide bond formation (Fig. 1a). A carbohydrate assay determined that Macroflor particles were composed of 40% polyglucose and 60% lysine. Optimizing chemical stoichiometry and reaction time yielded a mean particle diameter of $5.0 \pm 0.4$ nm (Fig. 1b), which is small enough to allow renal excretion. Chemical characterization of Macroflor documented 23 nmol free amines per mg and 62 nmol azides per mg, providing two strategies for derivatization with imaging beacons.

Free amines were capped by reaction with excess succinic anhydride, which shifted the zeta potential from $-11.1$ to $-19.9 \pm 1.0$ mV, thereby indicating a change in the surface charge. Succinylation did not change particle size. Macroflor was radiolabelled with [18]F with high radiochemical purity (Fig. 1c). The average specific activity was $10.2 \pm 1.9$ mCi mg$^{-1}$.

Macroflor was first tested in healthy mice (Fig. 1d,e) and rabbits (Fig. 1f). In both species, the highest activity occurred in the kidneys, while uptake in liver and spleen was considerably lower, as measured 120 min after injection. Together with the rapid clearance from the blood pool (Fig. 1d, blood half-life of 6.5 min measured in five normal mice and 22.5 min measured in four normal rabbits), these data indicate that Macroflor is rapidly cleared from circulation by the renal system. Macroflor stability in mouse serum was tested by size exclusion radiochromatography. The data indicate that Macroflor is stable for 120 min with no appearance of additional peaks (Supplementary Fig. 1).

To test if similarly rapid renal excretion occurs in primates, we conducted a dynamic imaging experiment using a clinical PET/ magnetic resonance imaging (MRI) scanner (Fig. 2a–c). Macroflor, 2.5 mCi, was injected intravenously, followed by 90 min of PET data acquisition. As anticipated, Macroflor rapidly departed the blood pool and accumulated in the kidneys and bladder (Fig. 2a,b). Fitting the blood pool activity yielded a blood half-life of 21.7 min (Fig. 2c). Thus, data obtained in all three species document rapid renal excretion of Macroflor.

To explore Macroflor uptake by macrophages, we used dynamic intravital confocal microscopy to study the hearts of $Cx_3cr_1^{GFP/+}$ mice. In naive hearts, no signal was seen in the 680 nm channel, but cardiac macrophages were brightly green fluorescent protein (GFP) positive (Fig. 2d). Shortly after the fluorescent nanoparticle was injected into the tail vein, intravascular signal appeared in cardiac vessels. While blood pool signal faded rapidly thereafter, we observed nanoparticle enrichment in cardiac macrophages (Fig. 2d,e). These data imply that the nanoparticles undergo rapid distribution to macrophages, and that these nanoparticles are cleared from the blood pool within a time frame that enables [18]F PET imaging of macrophages residing in cardiovascular tissues.

To test the hypothesis that Macrolite enters macrophages via a biologically active process, we studied *in vitro* uptake into primary cells harvested from mouse spleens at different temperatures. Splenocytes were incubated with the fluorescently labelled nanoparticle Macrolite for 15 min and 12 h at either 4 or 37 °C. Uptake of Macrolite was assessed by flow cytometry, which also identified macrophages by their typical surface marker profile. We found substantial Macrolite uptake only in the cells that were incubated at 37 °C, indicating that nanoparticle uptake is inhibited if the cells are on ice (Supplementary Fig. 2). These data indicate that Macrolite uptake is a biologically active process.

**Imaging inflammatory atherosclerosis.** Macrophages enter arterial wall segments to remove cholesterol deposits. In patients and animals with atherosclerosis, the cells fail in this task and instead create sub-endothelial inflammatory lesions. Continued recruitment and local proliferation feed macrophage population growth, which perpetuates inflammation and swells lesions that impede blood flow. Macrophages also destabilize arterial tissue integrity by secreting proteolytic and pro-inflammatory enzymes, which can lead to atherosclerotic plaque disruption, thrombotic arterial occlusion and sudden downstream ischemia. This makes macrophages a potential therapeutic target and indicates the need to monitor them. In $ApoE^{-/-}$ mice on pro-atherogenic diet, we observed Macroflor enrichment in the aortic root and arch (Fig. 3a,b), vascular territories that are affected by inflammatory

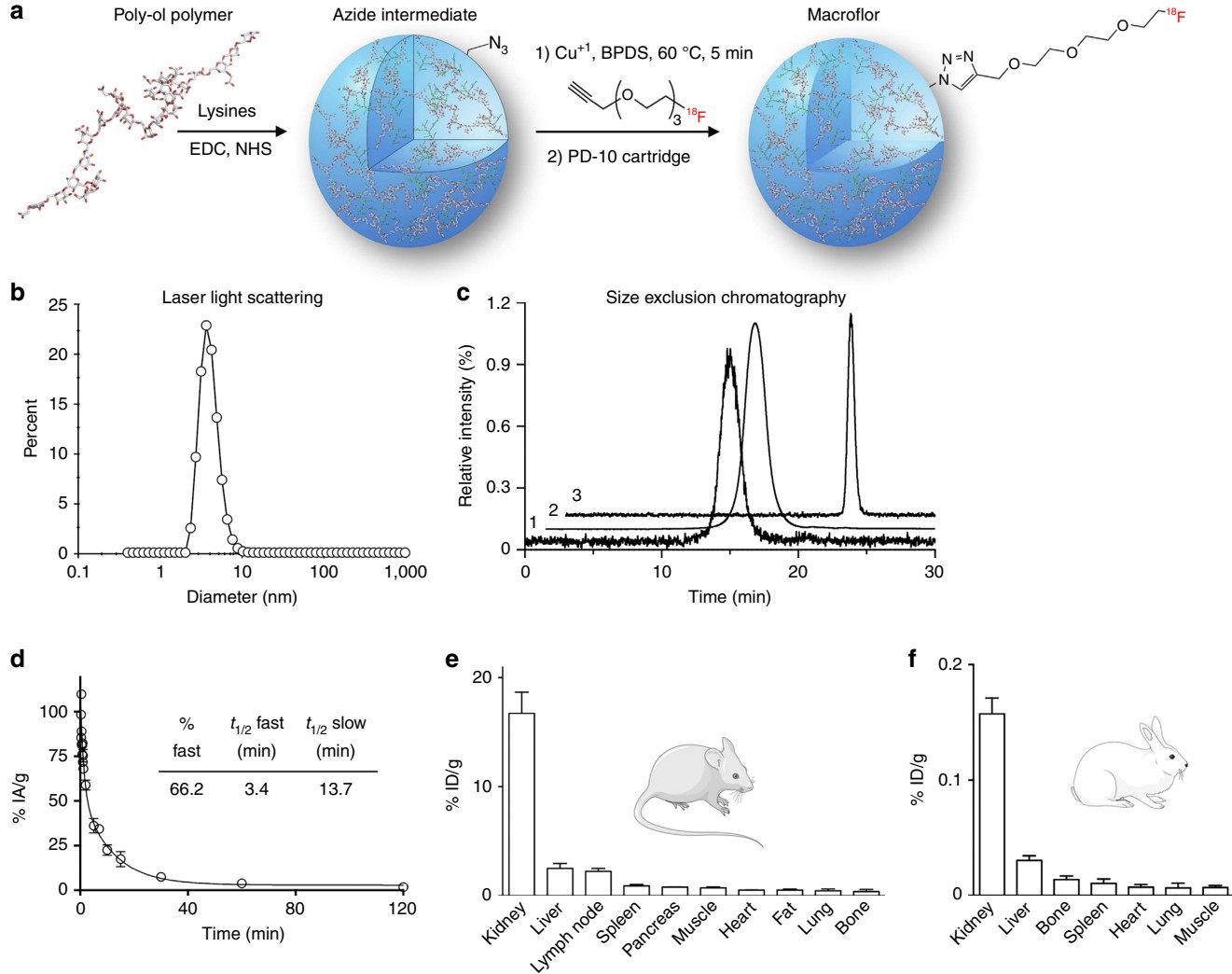

**Figure 1 | Chemistry and pharmacokinetics in mice and rabbits.** The figure describes synthesis, labelling, quality control and biodistribution in two species. (**a**) Synthesis and radiolabelling scheme. (**b**) Dynamic light scattering measurement of purified nanoparticle. (**c**) Size-exclusion chromatogram of purified labelled and unlabelled nanoparticles. (**d**) Blood half-life in wild-type mice ($n = 5$). (**e**) Biodistribution data in mice ($n = 5$) and (**f**) in normal rabbits ($n = 4$). Mouse and rabbit cartoon images are reproduced from the servier medical art image data bank (http://www.servier.com/Powerpoint-image-bank).

atherosclerosis. Standard uptake values obtained by PET/ computed tomography (CT) imaging 120 min after intravenous injection of $525 \pm 167 \,\mu$Ci Macroflor were significantly increased in mice with atherosclerosis (Fig. 3c). These *in vivo* data correlated well with activity measured in excised aortae (Fig. 3d,e). Using autoradiography exposure of aortae, we detected radioactivity co-localized with fatty atherosclerotic plaques (Fig. 3f). The nanoparticle's uptake profile into cells that reside in the aortic wall showed a significant predilection for macrophages, while other leukocytes such as lymphocytes and neutrophils showed negligible nanoparticle incorporation (Fig. 3g–i). Immunofluorescent labelling of the myeloid marker CD11b, which is highly expressed by macrophages, co-localized with nanoparticles in histological sections of aortic plaques (Fig. 3j).

We next explored Macroflor imaging of atherosclerosis in rabbits using a clinical PET/MRI scanner. Ninety minutes after Macroflor injection, we observed high PET signal in the rabbits' kidneys (Fig. 4a,b), thereby confirming Macroflor's renal excretion. Significantly increased PET signal localized into the infrarenal aortae of rabbits that developed atherosclerotic lesions therein after exposure to atherogenic diet and infrarenal aortic balloon injury (Fig. 4a,c). In a cohort of rabbits that received balloon injury with

intermediate pressure, aortic standard uptake values were higher than in the control cohort but lower than in rabbits that received full-pressure endothelial denudation (Fig. 4c). Autoradiography on excised aortae documented radioactivity in a pattern reflecting atherosclerotic lesion distribution (Fig. 4d). *Ex vivo* scintillation counting of excised rabbit aortae revealed increased activity in rabbits with atherosclerotic disease (Fig. 4f). Two days before Macroflor imaging, rabbits underwent [18]F-fluorodeoxyglucose ([18]F-FDG) PET to enable comparison. The rabbit cohorts had similar PET signal distributions (Fig. 4f). When aortic segments were studied, [18]F-FDG and [18]F-nanoparticle signal correlated with an $R^2 = 0.518$ ($P < 0.0001$, Pearson correlation coefficient, Fig. 4g). The difference in myocardial uptake between Macroflor and [18]F-FDG, shown in Fig. 4h and Supplementary Fig. 3 may have implications for cardiac PET inflammation imaging.

**PET imaging of macrophages in acute MI.** Ischaemic organ injury profoundly expands the macrophage pool. The cells derive from blood monocytes made in the bone marrow and spleen but also proliferate locally in the infarct[7]. Macrophage numbers peak around day 3 after ischemia, before tissues transition to resolution

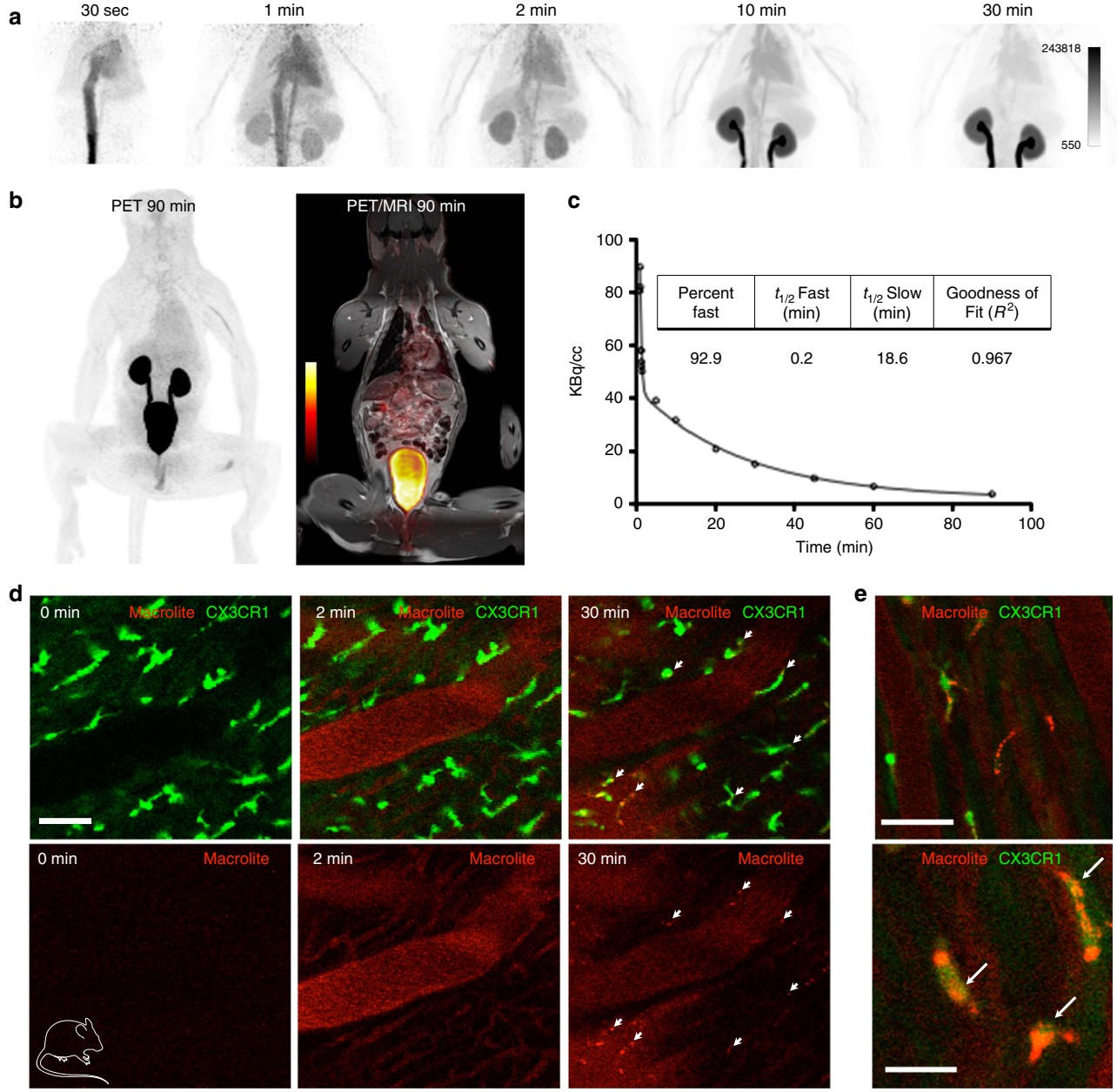

**Figure 2 | Macroflor PET/MRI in non-human primate and rapid uptake into mouse cardiac macrophages.** Macroflor is excreted rapidly via the kidneys and is taken up by cardiac macrophages. (**a**) PET maximum intensity projection images from dynamic scan. A single primate experiment was carried out. (**b**) Full-body PET/MR images 90 min into dynamic scan (Left: PET; Right: PET/MRI). (**c**) Macroflor time-activity curve derived from PET SUV data in non-human primate cardiac blood pool. (**d**) Intravital dynamic confocal microscopy of Macrolite uptake into cardiac macrophages carried out in a single Cx₃crf^GFP/+ reporter mouse. The upper panel shows green GFP signal indicating cardiac macrophages and red VT680 signal reporting on Macrolite presence. The 0 min image was acquired before fluorescent probe injection into the tail vein. Intravascular signal is detected 2 min later, while the 30 min time point illustrates co-localized Macrolite and GFP⁺ macrophages. Scale bar, 50 μm. (**e**) Higher magnification images illustrate co-localization of Macrolite and macrophage signal. Scale bars, 50 μm (upper panel) and 25 μm. Mouse cartoon image is reproduced from the servier medical art image data bank (http://www.servier.com/Powerpoint-image-bank).

of inflammation. During the resolution phase, macrophages assume less inflammatory phenotypes and promote tissue repair. However, their overabundance, or a delayed phenotype transition, promotes heart failure. Infarct macrophages are, therefore, a potential therapeutic target, but we currently lack tools for monitoring them quantitatively. To explore Macroflor PET imaging after MI, we imaged mice with permanent coronary ligation. *In vivo* imaging showed Macroflor uptake into the ischaemic myocardium (Fig. 5a,b) to be significantly higher than myocardial standard uptake values in control mice without MI

(Fig. 5c). These *in vivo* data were corroborated by scintillation counting (Fig. 5d) and autoradiography exposure (Fig. 5e) of excised mouse hearts. Flow cytometric evaluation of cardiac leukocytes showed significant enrichment in macrophages (Fig. 5f–h).

**Dual targeted PET/MRI reports on phenotype and number.** Macrophages may not only change in number but also shift their phenotypes and inflammatory activities. A change in

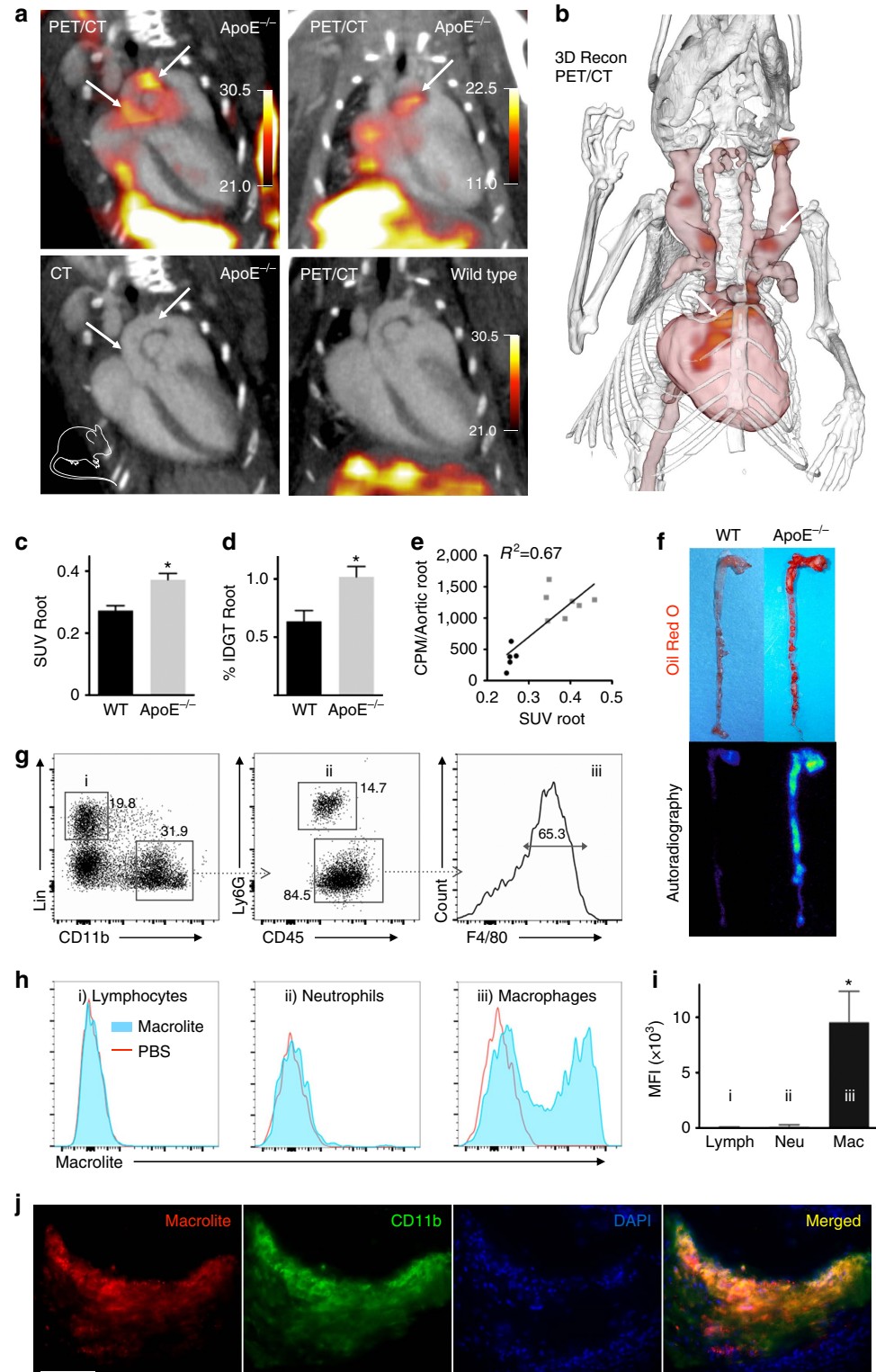

**Figure 3 | Macroflor PET/CT in mice with atherosclerosis.** Macroflor-enabled macrophage imaging in aortic plaques of mice with atherosclerosis. (**a**) Representative PET/CT images of several experiments in $ApoE^{-/-}$ and wild-type control mice after IV Macroflor injection. PET scale bar is in kBq/cc $n = 14$. (**b**) Three-dimensional rendering derived from PET/CT in $ApoE^{-/-}$ mouse shows PET signal in red (arrows). (**c**) *In vivo* standard uptake values (SUV) for aortic roots of wild-type and $ApoE^{-/-}$ mice ($n = 5$–7 per group, unpaired *t*-test). (**d**) *Ex vivo* gamma count reports percent injected dose per gram aortic tissue (%IDGT (percent injected dose per gram tissue), $n = 5$–7 per group, unpaired *t*-test). (**e**) Correlation of (**c,d**) for individual wild-type (black) and $ApoE^{-/-}$ mice (grey), counts per minute (CPM). (**f**) *Ex vivo* Oil-Red-O staining and corresponding autoradiography of representative aortae. (**g**) Flow cytometric gating on aortic cells after IV Macrolite injection. (**h**) Mean fluorescence intensity (MFI) of Macrolite in respective cells retrieved from $ApoE^{-/-}$ mouse aorta. (**i**) VT680 fluorescence indicating Macrolite uptake, obtained in three $ApoE^{-/-}$ mouse aortae (one-way analysis of variance). (**j**) Fluorescent microscopy of aortic root plaque after IV Macrolite injection. Scale bar, 100 μm. Data are shown as mean ± s.e.m., * indicates $P < 0.05$. Mouse cartoon image is reproduced from the servier medical art image data bank (http://www.servier.com/Powerpoint-image-bank).

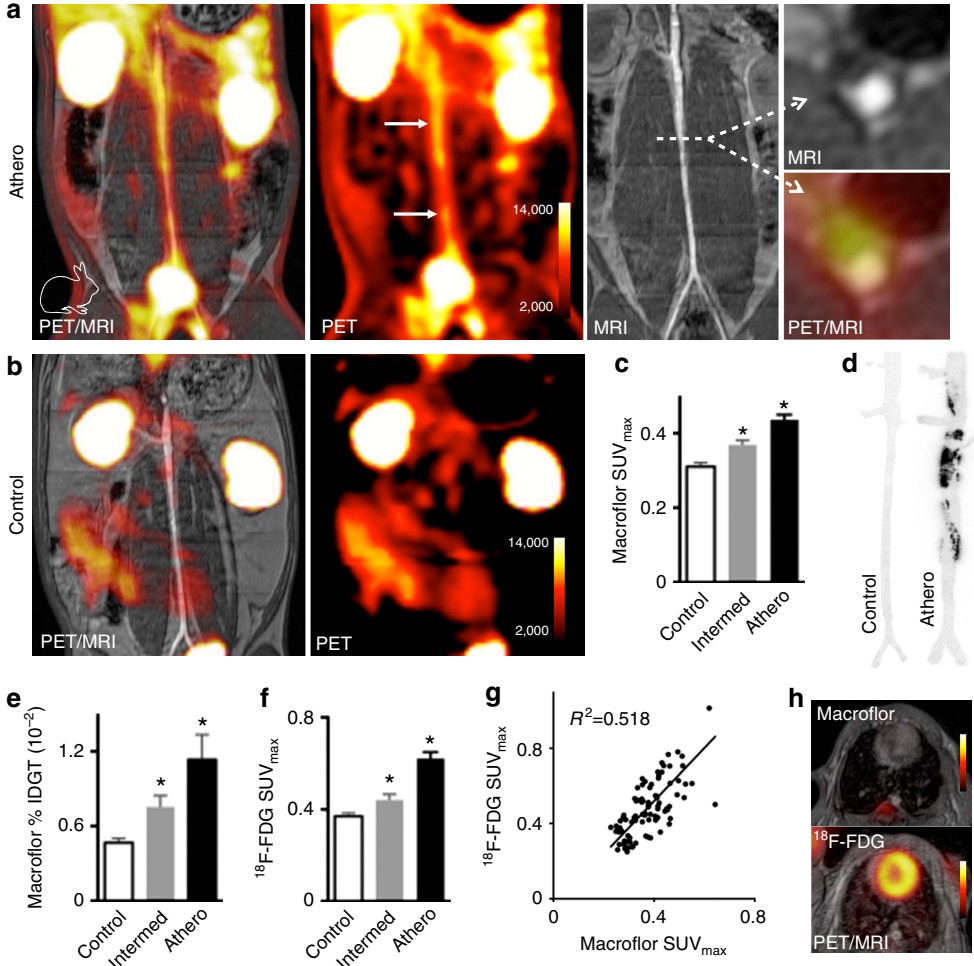

**Figure 4 | PET imaging of inflammatory atherosclerosis in rabbits.** Macroflor PET imaging detects atherosclerotic aortic plaques in rabbits, with distinct differences from [18]F-FDG. (**a**) PET/MR images obtained in rabbits with atherosclerosis and (**b**) control rabbits. (**c**) Standard uptake values (SUV) in infrarenal aorta after Macroflor injection in control rabbits, rabbits with intermediate and full aortic balloon infrarenal aorta denudation. Experiments were done in four rabbits per group, with analysis in 38–40 aortic regions (one-way analysis of variance (ANOVA)). (**d**) Autoradiography of the abdominal aorta in a control rabbit and a rabbit with atherosclerosis after Macroflor injection, representative images of 12 autoradiography exposures ($n = 4$ per group). (**e**) Ex vivo gamma counting reports percent injected dose per gram aortic tissue (%IDGT (percent injected dose per gram tissue), $n = 4$ per group, one-way ANOVA). (**f**) Two days before macrophage imaging with Macroflor, the same rabbits underwent [18]F-FDG PET imaging to report aortic SUV in infrarenal aorta ($n = 4$ rabbits per group, analysis in 23–38 aortic regions, one-way ANOVA). (**g**) Correlation of [18]F-FDG with Macroflor in vivo PET signal ($P < 0.001$, $n = 12$ rabbits). (**h**) Cardiac PET images with respective agents. Data are shown as mean ± s.e.m., * indicates $P < 0.05$. Rabbit cartoon image is reproduced from the servier medical art image data bank (http://www.servier.com/Powerpoint-image-bank).

macrophage phenotype occurs when infarct inflammation transitions to resolution on day 3 in wild-type mice with permanent coronary ligation. In mice and patients with inflammatory co-morbidities such as pre-existing athero-sclerosis, oversupply of inflammatory myeloid cells compromises resolution and leads to post-MI heart failure because inflammatory macrophage phenotypes impede tissue regeneration. Consequently, monitoring macrophage numbers and phenotypes is vital to furthering our understanding of ischaemic heart disease. Recently-developed PET/MRI systems can image more than one target simultaneously by combining two imaging agents detectable with either PET or MRI. We here combined Macroflor PET imaging with MRI sensing of myeloperoxidase (MPO), an enzyme expressed by inflammatory macrophages. We previously validated the gadolinium (Gd)-based, activatable enzyme reporter MPO-Gd for infarct imaging[8] and found the signal to be specific for inflammatory myeloid cells, especially neutrophils and the inflammatory L6C[high] monocyte subset[9].

We tested dual-target PET/MR imaging at two time points after acute MI. The first imaging occurred on day 2 after coronary ligation, which coincides with the inflammatory phase character-ized by an abundance of inflammatory neutrophils, monocytes and macrophage subsets (Fig. 6a). The second imaging session was on day 6 (Fig. 6b), which coincides with the resolution phase of infarct healing, when reparative macrophage phenotypes support tissue healing via crosstalk with fibroblasts and endothelial cells in sprouting neo-vessels. The PET standard uptake value for reporting macrophage numbers increased between day 2 and day 6 (Fig. 6c), a change that indicates an expanding macrophage pool in the healing infarct. The increase in Macroflor PET signal correlates well with previous flow cytometric studies documenting rising macrophage numbers between days 2 and 6 after MI[7,10]. The data also imply that macrophages of different phenotypes readily incorporate Macroflor. Concomitantly, we observed declining MRI MPO signal (Fig. 6d). Thus, while the macrophage population expanded, the cells produced less MPO, a dynamic that is

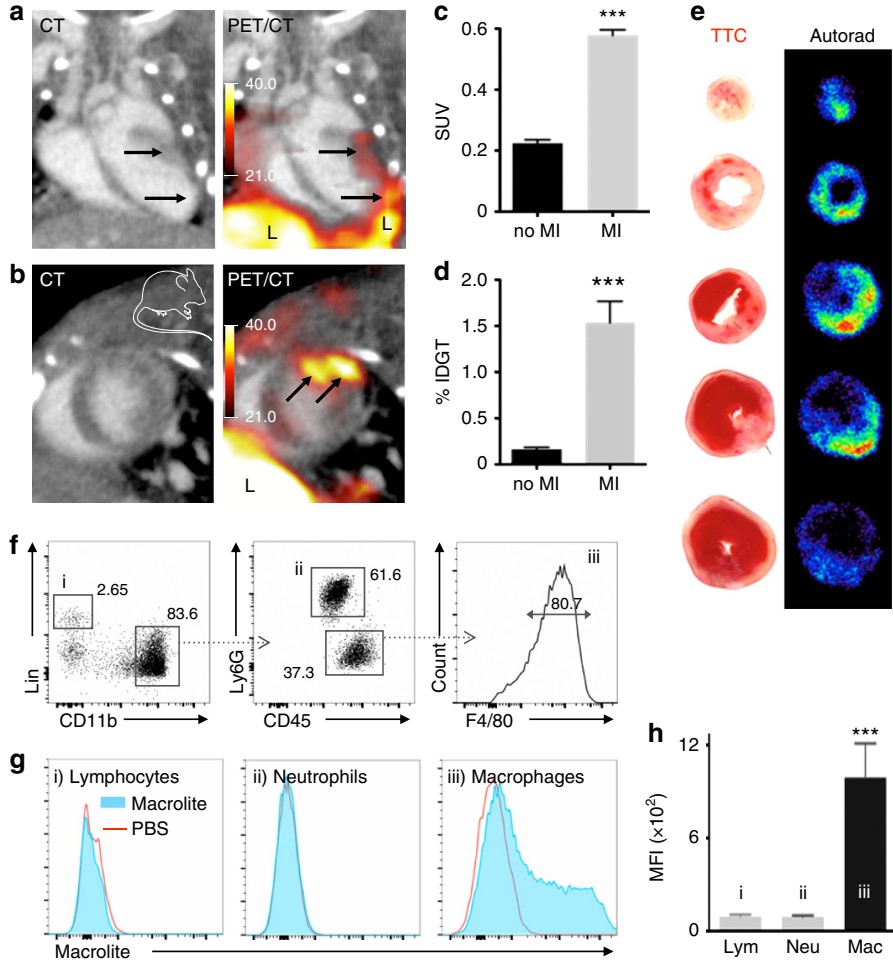

**Figure 5 | Macrophage PET imaging in murine myocardial infarction (MI).** Macroflor enriches in macrophages residing in acute infarcts. (**a**) PET-CT long axis and (**b**) short axis views following IV Macroflor injection into C57BL/6 mice on day 6 after MI. PET scale bar is in kBq/cc. (**c**) *In vivo* SUV quantification of Macroflor signal in non-infarcted and infarcted heart tissue ($n = 4$ per group, unpaired t-test). (**d**) *Ex vivo* gamma counting data of MI tissue (%IDGT (percent injected dose per gram tissue), $n = 4$–5, unpaired *t*-test). (**e**) *Ex vivo* autoradiography signal corresponds with pale triphenyltetrazolium staining of day 2 infarct on short axis sections. Representative images from experiments obtained in four mice with infarcts. (**f**) Flow cytometric gating on infarct leukocytes after IV Macrolite injection. (**g**) Histograms of Macrolite uptake into respective cell populations. (**h**) Mean fluorescence intensity (MFI) reflecting Macrolite uptake, obtained in three mouse infarcts (one-way analysis of variance test). Data are shown as mean ± s.e.m., * indicates $P < 0.05$, *** indicates $P < 0.0001$. Mouse cartoon image is reproduced from the servier medical art image data bank (http://www.servier.com/Powerpoint-image-bank).

consistent with infarct inflammation switching to resolution. This sensing strategy could identify at-risk individuals that do not undergo phenotype switching.

Finally, we tested dual-target PET/MRI in mouse athero-sclerosis. Figure 6e illustrates the activation and retainment of MPO-Gd in the aortic root, a site of inflammatory atherosclerosis in $ApoE^{-/-}$ mice consuming a pro-atherogenic diet. After MPO-Gd injection, the contrast-to-noise ratio rose significantly (Fig. 6f). Immunostaining for MPO demonstrated its presence in the region of interest (Fig. 6g). To provide contrast with the macrophage resolution phenotype observed in day 6 infarcts, we subjected $ApoE^{-/-}$ mice to coronary ligation, which promotes disease-exacerbating inflammation systemically and in plaques[11]. *Ex vivo* histology and flow cytometry previously documented that acute MI accelerates atherosclerosis in mice. Comparing litter mate $ApoE^{-/-}$ mice with and without MI, we detected not only increased Macroflor PET signal but also increased MPO-Gd retention by MRI. Increased signal in both imaging channels indicates that macrophage numbers expanded but without the resolution documented by lower MPO-Gd

retention in the 6 day old infarct; indeed, plaque inflammation flared. *Ex vivo* scintillation counting confirmed a higher Macroflor uptake in aortic specimens harvested from mice after MI (Fig. 6k). In the clinic, such an integrated imaging strategy may identify disease-promoting inflammatory incidents.

## Discussion

Imaging cardiovascular targets has specific challenges because the tissues are in direct contact with the blood pool in which the imaging probe circulates. Macrophages, cells with high patho-physiological relevance, internalize nanoparticles that can deliver isotopes, fluorochromes or rare earth metal to be detected by nuclear, optical and MRI. Among those modalities, PET is uniquely quantitative, and in the clinic, [18]F is the most commonly used isotope. Until now, slow hepatic nanoparticle elimination impeded PET macrophage imaging with [18]F labelled particles, as the isotope decay usually outpaces nanoparticle elimination from the blood pool. Here we describe how Macroflor, a polyglucose nanoparticle shrunk to a size below the renal excretion threshold,

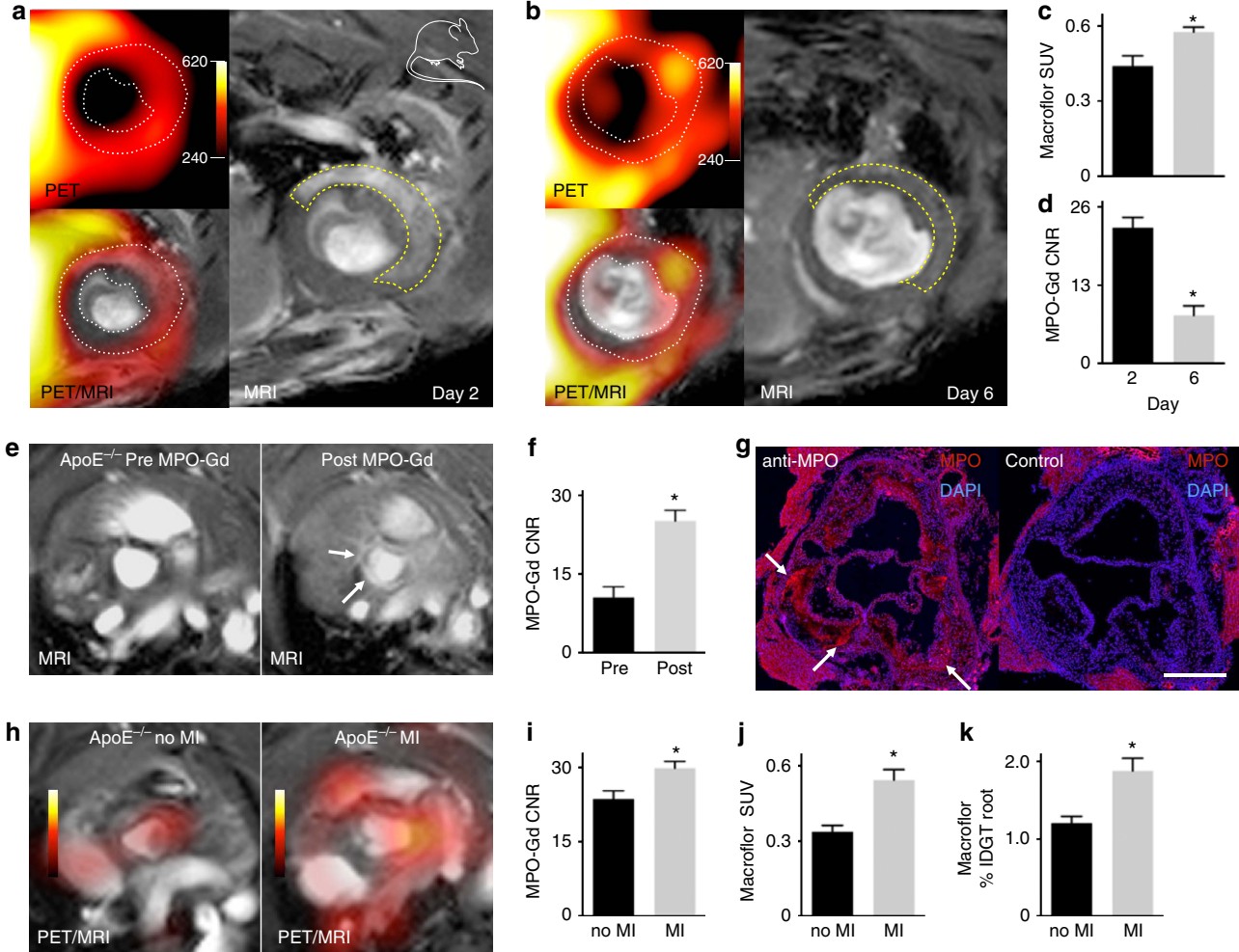

**Figure 6 | Dual channel macrophage PET/MRI in ischaemic heart disease.** Imaging macrophages and myeloperoxidase detects different inflammatory phenotypes. (**a**) PET/MRI on day 2 and (**b**) day 6 post MI in wild-type mice. White dotted line on PET/MRI outlines myocardium. Yellow dashed line on MRI outlines the infarct identified by gadolinium enhancement and wall motion abnormality in cine loops. (**c**) *In vivo* PET standard uptake value (SUV) in infarct zone on days 2 and 6 post MI ($n = 4$ per group, unpaired *t*-test). (**d**) *In vivo* MRI contrast to noise ratio (CNR) in infarct 90 min after IV MPO-Gd injection ($n = 4$ per group, unpaired *t*-test). (**e**) MRI of aortic root in $ApoE^{-/-}$ mouse before and 90 min after IV MPO-Gd. Arrows indicate enhancement in the aortic root. (**f**) Quantified MRI contrast to noise ratio (CNR) before and after MPO-Gd administration ($n = 8$ per group, unpaired *t*-test). (**g**) Immunofluorescent staining for the MR imaging target myeloperoxidase (MPO) in aortic root of $ApoE^{-/-}$ mouse. Control stain: omission of primary antibody on an adjacent slide. Scale bar, 500 μm. Representative image of $n = 4$ $ApoE^{-/-}$ mice. (**h**) PET/MRI following systemic administration of Macroflor and MPO-Gd in $ApoE^{-/-}$ mice without and 3 weeks after MI ($n = 4$-5). (**i**) MPO-Gd MRI contrast to noise ratio (CNR) in aortic roots of $ApoE^{-/-}$ mice with and without MI. (**j**) *In vivo* PET SUV of aortic root after IV Macroflor injection ($n = 4$-5 per group, unpaired *t*-test). (**k**) *Ex vivo* scintillation counting of aortae harvested from $ApoE^{-/-}$ mice after IV Macroflor injection ($n = 4$-5 per group, unpaired *t*-test). Data are shown as mean ± s.e.m., * indicates $P < 0.05$. Mouse cartoon image is reproduced from the servier medical art image data bank (http://www.servier.com/Powerpoint-image-bank).

solves this problem. Though it is excreted within minutes, Macroflor's high avidity for macrophages makes it a suitable PET agent for imaging these cells. Because Macroflor's synthesis relies on sugar molecules and uses facile click labelling for PET isotope attachment, it is highly suitable for clinical translation.

Sensitive imaging tools, including optical detection of fluorescent proteins expressed under control of macrophage-specific promoters, recently revealed macrophage presence and density in cardiovascular and other organs. In the steady state, these cells pursue surveillance and defense. One of their prime activities, phagocytosis of dying cells, infectious agents and other foreign material, also results in avid nanoparticle uptake. Macrophages expand by orders of magnitude in injured and diseased tissue, and depending on their numbers and inflammatory properties, macrophages may promote healing or disease. In cardiovascular

organs, overabundance of inflammatory macrophages damages vascular and cardiac structures, ultimately leading to ischemia, stroke, heart failure and death[3]. Thus, macrophages are being explored as therapeutic targets in heart disease and many other pathologies. But because macrophages also pursue salutary functions in tissue repair and defense, broadly targeting macrophages may have deleterious effects. As a consequence, imaging tools for monitoring macrophages or their functions will likely be companion strategies during drug development. Different imaging strategies for monitoring macrophages have been proposed, comprising imaging of adhesion molecules, cell surface receptors and secreted factors executing inflammatory functions[4]. These strategies likely differ in their sensitivity and specificity for macrophage presence, phenotype target range and pharmacokinetics. Some approaches may be sensitive for early

cell activation, such as imaging of alarmins[12], while others, including PET imaging of Macroflor enrichment, report on macrophages in all inflammatory stages, even on non-inflammatory tissue resident cells. We envision that Macroflor will enable detection of increased macrophage numbers in cardiovascular organs and monitor the cell population size as a function of therapy. Large animal infarct imaging is required next, since PET imaging in mouse thoracotomy infarct model reaches limitations in terms of spatially distinguishing inflammation in the body wall from myocardial injury. When combined with an MR imaging agent, dual-target data may additionally report on orthogonal biomarkers that reflect macrophage phenotypes.

Cardiovascular $^{18}$F-FDG PET detected arterial and myocardial inflammation in several preclinical studies[13–15] and clinical trials[16–22]. We, therefore, correlated $^{18}$F-FDG and Macroflor uptake in rabbits with atherosclerosis. The data show imaging agent overlap. In rabbits with atherosclerosis[13] and in human endarterectomy specimens[17], $^{18}$F-FDG associates with macrophages detected by histology, which supports the correlation we observed between Macroflor and $^{18}$F-FDG. However, the 0.518 correlation coefficient suggests that both agents behave distinctly. For example, if not sufficiently suppressed with appropriate measures, $^{18}$F-FDG enriches in cardiomyocytes, especially if these are distressed[23]. This consideration may be relevant for imaging the myocardium and coronary arteries. We observed negligible myocyte uptake of Macroflor, perhaps dictated by the different distribution and tissue penetration of small molecule and nanoparticle based PET imaging agents. The favourable Macroflor uptake profile for macrophages and its rapid pharmacokinetics motivate our next translational steps, which include imaging in large animals and toxicology in preparation for first-in-human studies.

## Methods

**Mice.** Female C57Bl/6 (B6) and apolipoprotein E knockout (ApoE$^{-/-}$) mice were purchased from the Jackson Laboratory (Jackson). ApoE$^{-/-}$ mice were on average age of 8–12-weeks-old when they began a high-cholesterol diet (Harlan Teklad, Madison, Wisconsin) for at least 12 weeks. MI was induced in both B6 and ApoE$^{-/-}$ mice by permanent coronary artery ligation. Anaesthetized with isoflurane, mice were intubated and ventilated. Left thoracotomy was performed in the fourth intercostal space to allow permanent ligation of the left anterior descending coronary artery with monofilament 8–0 suture (Ethicon, Somerville, NJ). The chest wall was closed with 7–0 nylon sutures and the skin was sealed with glue. Mice received isoflurane (2 to 3% v/v, Baxter) anaesthesia in all procedures. All animal experiments were approved by the Massachusetts General Hospital's Institutional Subcommittee on Research Animal Care.

**Rabbits.** Male New Zealand White rabbits (2.5–3 months old), purchased from Charles River Laboratories (Wilmington, MA) underwent double balloon injury of the thoracic and abdominal aorta to induce atherosclerosis. Denudation was performed by introducing a 4F-Fogarty embolectomy catheter (Edwards Lifesciences, Irvine, CA) under fluoroscopic guidance using a Philips Allura Xper FD20/10, Philips Healthcare (Best, The Netherlands). The catheter was introduced into the femoral artery, and the balloon was inflated to either 1 atm or 2 atm, depending on the degree of atherosclerosis desired. Surgery was performed under anaesthesia with intramuscular Ketamine (35 mg kg$^{-1}$) and Xylazine (5 mg kg$^{-1}$) injection. To further accelerate plaque progression, animals were fed a high cholesterol diet (Research Diets) enriched initially with 0.3% for 8 weeks, and subsequently 0.15% for 8 weeks, and was continued until study termination approximately 4 months after diet first began. The procedure was performed 2 weeks after start of high-cholesterol diet and repeated on the contralateral leg 4 weeks later. Male New Zealand White rabbits fed a standard chow diet served as controls. Experiments were performed in accordance with protocols approved by the Institutional Animal Care and Use Committees of the Icahn School of Medicine at Mount Sinai, NYC.

**Chemistry.** Unless otherwise noted, solvents and reagents were purchased from Sigma-Aldrich (St Louis, MO, USA) and used without further purification. Unless otherwise noted, water used for experiments and high-performance liquid chromatography (HPLC) was purified using a MilliQ filtration system (Waters). [$^{18}$F]-Fluoride ion (n.c.a.) in $^{18}$O-enriched water was purchased from PETNET

(Woburn, MA, USA). Analytical HPLC of radiolabelled compounds was performed with an Agilent 1,200 Series HPLC and a Poroshell 120 EC-C18 (4.6 × 50 mm 2.7 mm) reversed-phase column (Method C: eluents 0.1% formic acid (v/v) in H$_2$O (A) and MeCN (B); gradient: 0–0.3 min, 5% B; 0.3–7.5 min, 5–100% B; 7.5–10 min, 100% B; 2.5 ml min$^{-1}$) with a multichannel-wavelength ultraviolet/Vis detector, fluorescence detector and a flow-through γ-detector connected in series. Solid-phase extraction cartridges were Lichrolut C18 3-cc cartridge (200 mg, 30 mm particle size; Thermo, USA). Dynamic light scattering measurements to determine particle mean diameter and size distribution were performed on Zetasizer ZS and Zetasizer APS instruments (Malvern Instruments) in MilliQ water. Zetapotential measurements were performed on a Zetasizer ZS instrument (Malvern Instruments) in MilliQ water and PBS after performing calibration measurements on a commercial standard. Size-exclusion chromatography (SEC) was performed at a flow rate of 0.5 ml min$^{-1}$ using a Superdex 200 Increase (10 × 300) column (GE Healthcare) connected to an Agilent 1,200 Series HPLC with a multichannel-wavelength ultraviolet/Vis detector, fluorescence detector and a flow-through γ-detector connected in series. Mobile-phase for SEC was commercial 1 × PBS (Boston Bioproducts). Radio thin-layered chromatography (Radio-TLC) was performed on instant thin-layer chromatography (ITLC)-SG paper using 100% MeCN as mobile-phase. ITLC plates were read using a Bioscan AR-2000 radio-TLC scanner operated via a WinScan V3 software package.

**Macroflor synthesis.** L-Lysine (384 mg, 2.63 mmol), carboxymethylated polyglucose (22 chains; 550 mg, 2.19 mmol), azido-acetic N-hydroxysuccinimidyl (NHS) ester (38 mg, 0.19 mmol dissolved in 320 μl dimethylsulphoxide (DMSO)), propane sultone (320 μl, 0.16 mmol, 500 mM in DMSO), EDC (1.32 g, 7.19 mmol) and NHS (320 mg, 2.78 mmol) were combined and dissolved in 10 ml MES buffer (50 mM, pH 6.0) and stirred for 3 h at room temperature. The reaction product was isolated by pouring the mixture into ethanol and then pelletized by centrifugation. The pellet was dissolved in H$_2$O (MilliQ) and filtered through 0.22 μm spin filters. This crude product was purified by SEC. Purified polyglucose nanoparticles were washed repeatedly with H$_2$O (MilliQ) to remove phosphate buffer and lyophilized. Recovered weights of the fractionated nanoparticles were calculated, and polyglucose nanoparticles were dissolved in H$_2$O (MilliQ) to a concentration of 10 mg ml$^{-1}$. These solutions were used for physical (DLS size and zeta potential measurements) and chemical (quantifying sugar[24] and amines[25]) characterization. Reactive azides per mg macroflor were quantified in triplicate, by mixing Macrin (2.5 μl, 10 mg ml$^{-1}$) with tetrakis(acetonitrile)copper(I) hexafluorophosphate (20 μl, 80 mM in MeCN), bathophenanthroline-disulfonic acid disodium salt (20 μl, 80 mM in 1xPBS) and 5-propargyl-fluorescein (20 μl, 25 mM in dimethylformamide (DMF)) and flushing with argon for 1 min. This mixture was heated by microwave (60 °C, 30 W) for 5 min and was loaded onto a PD-10 column (conditioned with MilliQ water) followed by elution with MilliQ water. Fractions staining positive for nanopartiles were transferred to 10-kDa mwco filters and concentrated by centrifugation. Ultraviolet absorption measurements made at 490 nm (Nanodrop) were used to quantify the amount of 5-propargyl-fluorescein conjugated to nanoparticles using the Beer-Lambert equation, ($A = \varepsilon bc$, where $A$ is the absorbance, $\varepsilon$ is the molar absorbtivity (80,000 l mol$^{-1}$ cm$^{-1}$ for 5-propargyl-fluorescein), $b$ is the pathlength (cm) and $c$ is the concentration (mol l$^{-1}$) of 5-propargyl-fluorescein). In a control reaction, all reagents were added, except tetrakis(acetonitrile)copper(I) hexafluorophosphate, to test nonspecific binding of the fluorochrome to macrin. We found no nonspecific binding of 5-propargyl-fluorescein to macrin. For succinylation, macrin (200 μl, 10 mg ml$^{-1}$) was diluted with MES buffer (200 μl, 50 mM, pH 6.0), Et$_3$N (2 μl) and succinic anhydride (72 μl, 750 mM in DMSO). After shaking at 900 r.p.m. for 18 h at room temperature the reaction was loaded onto a PD-10 cartridge (conditioned with MilliQ water) and eluted with MilliQ water (2 × 1,000 μl followed by 8 × 500 μl fractions). The fractions were analysed for glucose content by spotting on a silica get TLC plate, developing with 5% H$_2$SO$_4$ in ethanol and then heating the plate. Fractions 3–7 spotted positive for glucose, and were combined and concentrated using 10-kDa mwco filters. The contents of the filters was washed with water (3 × 400 μl) resulting in 135 μl of concentrate.

**$^{18}$F nanoparticle labelling.** Nanoparticle $^{18}$F labelling used a bioorthogonal click reaction[26,27]. Briefly, [$^{18}$F]-fluoride, n.c.a., (~3,030 MBq, 82 ± 9 mCi) in H$_2^{18}$O (~300 μl), K$_2$CO$_3$ (350 μl, 33 mM) in water and Kryptofix 2.2.2 (350 μl, 33 mM, K222) in MeCN were added to a 10-ml microwave tube. The [$^{18}$F]-fluoride/K$_2$CO$_3$/ K222 mixture was dried by azeotropic distillation of water with MeCN. The tube was heated to 98 °C by microwave under a flow of argon, and 3 min later 1 ml of MeCN was added. At 7 and 11 min after the start of drying 1 ml of MeCN was added. The dried $^{18}$F/K$_2$CO$_3$/K222 was cooled to room temperature, 2-(2-(2-(prop-2-ynyloxy)ethoxy)ethoxy)ethyl tosylate in DMSO (400 μl) was added and the reaction vessel was sealed and heated to 90 °C for 10 min. After cooling to room temperature, the mixture was diluted with water (900 μl) and subjected to preparative HPLC purification on a Machery-Nagel Nucleodur C18 Pyramid 250 × 10 mm Vario-Prep column (5.5 ml min$^{-1}$ 30/70 MeCN/H$_2$O with 0.07% formic acid) with a 254 nm ultraviolet detector and radiodetector connected in series. [$^{18}$F]-3-(2-(2-(2-Fluoroethoxy)ethoxy)ethoxy)prop-1-yne ($^{18}$F-P3-C#C) was collected (tR = 8.1 min) in 5–6 ml of solvent and isolated by C18 solid-phase extraction. Elution (CH2Cl2, 600 μl) of the trapped material followed by

evaporation of organic solvent resulted in $30 \pm 4$ mCi of 18F-P3-C#C, a $52 \pm 8\%$ average decay-corrected radiochemical yield (dcRCY) in $56 \pm 4$ min ($n = 13$). Analytical HPLC demonstrated $>99\%$ radiochemical purity of $^{18}$F-P3-C#C. To the concentrated 18F-P3-C#C solution (100–200 µl of H2O), tetrakis(acetonitrile) copper(I) hexafluorophosphate (($Cu(CH_3CN)_4PF6$), 40 µl, 80 mM in MeCN), bathophenanthroline disulfonate (40 µl, 80 mM in $1 \times$ PBS) and Macrin-N3 (20 µl, $0.9 \pm 0.1$ mg, 5.5 mM azide in H2O) were added in a 1.5-ml centrifuge tube with a magnetic stir bar. The tube was flushed with argon for 30 s and closed. The sealed centrifuge tube was inserted into a 10-ml microwave test tube containing 1 ml H2O and heated to 60 °C (30 Watts) for 5 min. The centrifuge tube was removed, and the reaction mixture was analysed by radio-TLC (ITLC, 100% MeCN mobile phase). The reaction mixture was loaded onto a PD-10 (GE Healthcare, preconditioned with H2O) and eluted with H2O ($2 \times 1,000$ µl followed by $8 \times 500$ µl fractions). Fractions 4–7 were combined and concentrated using 10-kDa mwco filters resulting in $12 \pm 3$ mCi, a $37 \pm 7\%$ average dcRCY over the two-step synthesis in $109 \pm 8$ min ($n = 13$).

**Hot cell $^{18}$F nanoparticle labelling.** [$^{18}$F]-fluoride, n.c.a., ($\sim 37$ GBq, $1 \pm 0.3$ Ci) in $H_2^{18}O$ ($\sim 3,000$ µl) was loaded onto a quaternary methyl ammonium (QMA) cartridge and eluted with a solution containing $K_2CO_3$ (800 µl,150 mM) and Kryptofix 2.2.2 (9 mg, K222) in MeCN (1.92 ml) into a 10-ml sealed conical reaction vessel. Water was removed by azeotropic distillation at 120 °C in a heating block (10 min incubation) under vacuum and nitrogen-controlled stream flow. The dried $^{18}$F/$K_2CO_3$/K222 was cooled to room temperature and 2-(2-(2-(prop-2-ynyloxy)ethoxy)ethoxy)ethyl tosylate (5 µl, 5.6 mg) in DMSO (400 µl) was added. Following the reaction, the vessel was heated to 90 °C for 10 min. After cooling to room temperature (3 min), the mixture was diluted with water (900 µl) and purified by semi-preparative HPLC equipped with a reversed-phase C18 Luna $250 \times 10$ mm column (Phenomenex, Torrance, CA) in isocratic conditions (5.5 ml min$^{-1}$ 30/70 MeCN/H2O with 0.07% formic acid). [$^{18}$F]-3-(2-(2-(2-Fluoroethoxy)ethoxy)ethoxy)prop-1-yne ($^{18}$F-P3-C#C) was collected (tR = 7.1 min) in $\sim 5.5$ ml of solvent and isolated by C18 Lichrolut EN SPE (Millipore, Billerica, MA). The trapped material was further eluted with acetonitrile (1,750 µl) and the organic solvent dried at 65 °C resulting in a 23% average dcRCY [$^{18}$F]-P3-C#C in $45 \pm 5$ min ($n = 6$). Analytical HPLC demonstrated $>99\%$ radiochemical purity of $^{18}$F-P3-C#C. To the concentrated [$^{18}$F]-P3-C#C solution ($\sim 100$ µl of total volume) was added tetrakis(acetonitrile) copper(I) hexafluorophosphate (($Cu(CH_3CN)_4PF6$), 40 µl, 80 mM in MeCN), bathophenanthroline disulfonate (40 µl, 80 mM in $1 \times$ PBS), and Macrin-N3 (20 µl, 31.5 mg µl$^{-1}$, 156 nmol azide in H2O) in a 5-ml sealed glass vial with a magnetic stirring bar. The vial was flushed with nitrogen for 30 s, and the sealed vial was heated to 65 °C and reacted for 12 min. The reaction mixture was analysed by radio-TLC (ITLC, 100% MeCN mobile phase). The reaction mixture was loaded onto a PD-10 (GE Healthcare, preconditioned with H2O) and eluted with H2O ($2 \times 1,000$ µl followed by $4 \times 1,000$ µl fractions). Fractions were combined to produce a final $70 \pm 10$ mCi, a $13 \pm 4\%$ dcRCY over the two-step synthesis in $102 \pm 7$ min.

**Fluorochrome labelling.** In order to perform correlative fluorescence measurements, we also prepared a fluorescent version (Macrolite). In a 1.5-ml centrifuge tube, 100 µl macrin was diluted with MES buffer (200 µl, 50 mM, pH 6.0) and treated with Et3N (1.5 µl) and VT680XL-NHS (18.5 µl, 46.4 nmol, 2.5 mM in DMF). This combination was loaded onto a PD-10 cartridge (conditioned with MilliQ water) and eluted with MilliQ water ($2 \times 1,000$ µl followed by $8 \times 500$ µl fractions). Nanoparticle- and fluorochrome-positive fractions were concentrated using 10-kDa filters and characterized by Nanodrop and fluorescence measurements.

**Blood half-life and biodistribution measurements.** The blood half-life of Macroflor was determined with serial retro-orbital bleeds after tail vein injection of 100 µCi into five mice. Imaging agent biodistribution was analysed in a second cohort of mice killed 2 h after injection. Mice were perfused with 20 ml of saline. Organs were harvested and weighed, and their activity was measured with a gamma counter (1,480 Wizard 3-inch, PerkinElmer). Biodistribution data were corrected for decay, residual activity at the injection site, renal excretion and body weight.

**Stability of Macroflor in mouse serum.** Macroflor was incubated in mouse serum at 37 °C, and aliquots at different time points were analysed by SEC. Specifically, in $4 \times 0.5$-ml centrifuge tubes, Macrophor (5 µl, $27.3 \pm 0.3$ µCi) in $1 \times$ PBS was added to mouse serum (20 µl). One sample was immediately injected for SEC analysis while the other three tubes were capped and agitated at 37 °C. At 30, 60 and 120 min each sample was analysed by SEC. The resulting chromatograms were normalized for relative intensity.

**PET/CT mice.** Two hours after tail-vein injection of Macroflor ($525 \pm 167$ µCi in $120 \pm 10$ µl), PET/CT imaging was initiated in conjunction with high-resolution contrast-enhanced vascular CT (Inveon, Siemens). The PET data reconstruction was done using ordered subsets expectation maximization and filtered back

projection algorithms to achieve a spatial resolution approaching approximately 1 mm. To quantitate PET, anatomic CT data provided the basis for placing regions of interest (ROI). The CT images were acquired with 80 kVp and 500 µA X-ray power, 370 to 400 ms exposure time and 90 µm isotropic resolution.

**PET/MRI mice.** Mouse PET/CT and MRI were performed sequentially using a custom-designed mouse bed and PET/CT gantry adapter[14]. To detect the presence of MPO in MI and atherosclerotic lesions, MPO-Gd, a small-molecule gd-based activatable sensor for MR imaging of MPO activity[28], was injected IV (0.3 mmol kg$^{-1}$) 90 min before the MRI scan. Two to three cine image slices were obtained at mid-ventricular level on a 7 Tesla Pharmascan (Bruker) with a cardiac volume coil (Rapid Biomedical), electrocardiogram and respiratory gating (SA instruments) and a fast gradient echo FLASH sequence with the following parameters: echo time 2.9 ms; 16 frames per R-R interval (repetition time 14 ms); resolution 156 µm $\times$ 156 µm $\times$ 1 mm; number of excitations 4; flip angle 60 degrees. Contrast to noise ratios (CNR) were calculated for infarcts[8] as $(MSI_{MI\ area} - MSI_{Non-infarcted\ myocardium})/(SD_{Noise})$; for atherosclerotic lesions $CNR = (MSI_{Aortic\ root} - MSI_{Skeletal\ muscle})/(SD_{Noise})$; MSI, mean signal intensity; SD, standard deviation. Offline PET/CT and MRI data fusion was based on external fiducial markers. A custom-built mouse vest comprising several PE50 tubing loops was filled with 15% iodine in water, which is visible in both CT and MRI. PET data were fused to CT as part of a standard workflow. MRI and PET data set fusion was then obtained by superimposing the fiducial landmarks with AMIRA software (Version 5.4) and the open source software Osirix, as described before[14].

**PET/MRI rabbits.** Rabbits ($n = 12$, 8 atherosclerotic (weight: $3.4 \pm 0.4$ kg) and 4 healthy controls (weight: $3.5 \pm 0.7$ kg)) were injected with $^{18}$F-FDG ($5.23 \pm 0.73$ mCi) via the marginal ear vein 2 days before Macroflor PET imaging. Animals were fasted for 4 h before $^{18}$F-FDG administration. The radiotracer was allowed to circulate for 3 h, at which point a static scan was performed. Forty-eight hours after the $^{18}$F-FDG injection, rabbits were injected with Macroflor ($4.07 \pm 0.19$ mCi in approx. 2.5 ml), which circulated for 2 h and 50 min before a static PET imaging scan. PET/MR imaging was performed on a Siemens mMR 3 T PET/MR scanner using a body matrix and spine coil. After scout scans, the PET scan was initiated and acquired simultaneously with MRI sequences for vessel wall characterization. A bright-blood, three-dimensional (3D) time-of-flight non-contrast enhanced angiography sequence was acquired to localize arterial anatomical landmarks (renal arteries and iliac bifurcation). Imaging parameters were: TR, 23 ms; TE, 2.8 ms; flip angle, 20 degrees; spatial resolution, $0.7 \times 0.7 \times 1$ mm$^3$. PET images were reconstructed offline using the Siemens e7tools software package, interfaced with custom built Matlab software (http://www.mathworks.com). Attenuation correction was performed by segmenting images into 2 compartments (soft tissue and air). For pharmacokinetic analysis, blood was sampled via both central ear arteries at 1, 5, 10, 15, 30, 60, 120, 180 and 210 min post injection. Blood was weighed and its radioactivity content measured on a Wizard 2,480 automatic gamma counter (Perkin Elmer). Radioactivity distribution measurements were performed 20 min after PET scan completion (210 min after injection of the tracer). Rabbits were killed and perfused with 500 ml saline. Organs were excised, blotted and diced before weighing and counting. Digital autoradiography was performed by placing the aortic samples in a film cassette against a phosphor imaging plate (BASMS-2325, Fujifilm,) for 13 h at $-20$ °C. Phosphor imaging plates were read at 25 µm pixel resolution with a Typhoon 7000IP plate reader (GE Healthcare). Image analysis was conducted, using OsiriX Imaging Software, by drawing ROIs on the selected tissues (liver, kidneys, spleen and abdominal aorta from left renal artery to iliac bifurcation). Blood activity was quantified in the cardiac chambers. Standardized uptake values (SUVs, defined as (Pixel value (Bq per ml)*Weight of the subject (kg) per Dose (Bq))*1,000 g kg$^{-1}$) were obtained by averaging SUV$_{max}$ values in each ROI drawn on all slices over the whole organ or over at least 10 slices of the tissue of interest.

**PET/MRI baboon.** A male baboon (*Papio anubis*, weight = 22.6 kg) was deprived of food for 12 h before the study. Anaesthesia was induced with intramuscular ketamine (10 mg kg$^{-1}$) and xylazine (0.5 mg kg$^{-1}$). After endotracheal intubation, the baboon was catheterized antecubitally for radiotracer injection. Anaesthesia was maintained using isoflurane (1 – 1.5%, 100% oxygen, 1 l min$^{-1}$) during the scan, and ketamine/xylazine effects were reversed with yobine (0.11 mg kg$^{-1}$, intramuscular) before image acquisition. Vital signs, including heart rate, respiration rate, blood pressure, O2 saturation, and end tidal CO2, were monitored continuously and recorded every 15 min. Simultaneous PET/MR data were acquired using a Siemens Biograph mMR system (Siemens Healthcare, Erlangen, Germany). MR body imaging was performed with real-time respiratory bellow gating and using the body matrix coil and the built-in spine coil as the receiving coil elements. High-resolution anatomical T1-weighted turbo spin echo sequence was acquired with the following parameters: TR = 500 ms, TE = 9.5 ms, matrix size $384 \times 276$, FOV = 45 cm, phase FOV 68.8%, 5 mm slice thickness, and 35 slices. For MR-based attenuation correction of PET data, a T1-weighted, 2-point Dixon 3D volumetric interpolated breath-hold examination scan was obtained. PET data were obtained using a single-bed acquisition with an axial field of view of 25.8 cm, transverse field of view of 59.4 cm. PET data were acquired dynamically for 90 min

after 2.73 mCi Macroflor. PET data were stored in list mode, and reconstruction was performed using a 3D-OSEM method with detector efficiency, decay, dead time, attenuation and scatter corrections applied. Dynamic PET data were reconstructed into the following time-framing: $4 \times 15$ s, $2 \times 30$ s, $3 \times 1$ min, $5 \times 5$ min and $6 \times 10$ min. Dynamic whole-body PET images at different time points (30 s, 1 min, 2 min, 10 min and 30 min post-radiotracer injection) and a sum image over 90 min were shown as maximum image projection. ROIs were manually delineated from the T1-weighted anatomical image of the cardiac blood pool using PMOD 3.4 (PMOD Technologies Ltd., Zurich, Switzerland) to plot time-activity curves of the cardiac blood pool.

**Intravital microscopy.** Intravital microscopy was performed using a custom-designed system and imaging protocol developed and validated previously[29,30]. Cx3cr1$^{GFP/-}$ mice were sedated using inhaled 2% isoflurane mixed with 100% oxygen and intubated using a 22-gauge angiocath as an endotracheal tube. Volume mode ventilation was performed with a tidal volume of 0.25 cc and a rate of 130 breaths per minute, and animal body temperature was maintained at 37 °C using a heat plate. A left thoracotomy was performed in the fourth intercostal space, and the ventricular surface was exposed by gently separating the parietal pericardium. A custom-designed tissue stabilizer was affixed to the heart to minimize motion artifacts. Images were acquired using prospective real-time cardiac gating achieved through cardiac pacing[30]. Macrolite was administered via tail-vein injection during confocal microscopy that continued to acquire data at serial time points after injection. Simultaneous excitation with 488 and 633 nm and two-channel detection for GFP and VT680 was performed, and composite two-colour images were created by merging the detection channels in RGB space.

**Flow cytometry.** Two days after they received infarcts, C57BL/6 mice (age 12 weeks, $n = 3$) and ApoE$^{-/-}$ mice (HCD diet 12–16 weeks, $n = 3$) received IV injection of Macrolite (5 nmol VT680) and were euthanized 2 h later for flow cytometry analysis. All animals were perfused from the left ventricle with 20 ml PBS at 4 °C. Infarcted heart segments from C57BL/6 mice were harvested. ApoE$^{-/-}$ mouse aorta were excised using a dissecting microscope. Infarcted myocardial and aortae were minced and digested at 37 °C for 1 h with agitation at 750 r.p.m. in medium containing collagenase I (450 U ml$^{-1}$), collagenase XI (125 U ml$^{-1}$), DNase I (50 U ml$^{-1}$) and hyaluronidase (60 U ml$^{-1}$) (all from Sigma-Aldrich). The tissues were then passed through 40 μm cell strainers and re-suspended in PBS with 0.5% bovine serum albumin (FACS buffer). The processed single cell suspensions (300 μl) were stained with a biotin-conjugated anti-mouse lineage monoclonal antibody cocktail, containing antibodies directed against CD90 (30-H12), B220 (RA3-6B2), NK1.1 (PK136) and Ter-119 (TER-119) (from either Biolegend or BD Biosciences/Pharmigen), used in combination with Streptavidin-Pacific Orange (Life Technologies). Myeloid cells were then stained with anti-mouse CD45 (104), CD11b (M1/70), F4/80 (BM8) and Ly-6G (1A8). Macrophages were defined as CD45$^+$, Lineage$^-$ (CD90/B220/NK1.1/Ter-119)$^-$ CD11b$^+$ Ly6G$^-$ F4/80$^{high}$, neutrophils as CD45$^+$ Lineage$^-$ CD11b$^+$ Ly6-G$^+$ and lymphoid lineage cells as CD45$^+$ Lineage$^+$. Flow cytometry was performed on a LSR II flow cytometer (BD Biosciences). Data were analysed with use of FlowJo software (Tree Star).

**Oil-Red-O staining and autoradiography.** Oil-Red-O staining depicted the distribution of plaques in ApoE$^{-/-}$ aortae that were subsequently photographed by a digital flatbed scanner and analysed by digital autoradiography. Aortae were exposed to a phosphor imager plate and read with a Typhoon FLA9000 system (GE Healthcare) 12 h later. To stain for lipid contents in atherosclerosis, aortae were briefly incubated in 60% 2-propanol and stained with 0.5% Oil-Red-O solution for 15 min at room temperature. After rinsing in 60% 2-propanol, aortae were washed repeatedly in PBS. Scanned autoradiography images were visualized using the ImageJ program (1.440).

**Triphenyltetrazolium staining.** MI was visualized after freshly produced myocardial rings were incubated with 1% triphenyltetrazolium (Sigma Aldrich) in PBS. Tissue was subsequently recorded with a digital flatbed scanner. Autoradiography was done by exposure on the PhosphorImager for ~12 h.

**Histology.** To analyse microscopic imaging agent distribution in atherosclerotic lesions, aortic roots were excised from ApoE$^{-/-}$ mice 2 h after IV Macrolite injection, embedded in OCT compound (Sakura Finetek) and flash-frozen in a 2-methylbutane bath with dry ice. Ten micrometre thick serial sections were prepared and stained with FITC-CD11b (clone: M1/70, BD Biosciences, dilution 1:25). The sections were counterstained with 4′,6-diamidino-2-phenylindole, dihydrochloride (Life Technologies, dilution 1:3000), and fluorescence microscopy was performed using a Olympus BX63 fluorescence microscope equipped with an ANDOR Neo sCMOS Monochrome Camera. To detect the presence of MPO in atherosclerotic lesions, aortic roots were harvested from ApoE$^{-/-}$ mice after MRI, embedded and flash-frozen. The 6 μm sections were stained for MPO (clone: Ab-1, Thermo Scientific, dilution 1:50), and the control staining omitted the primary antibody. Biotinylated anti-rabbit IgG (Vector Labs dilution 1:100) and streptavidin

DyLight 594 (Vector Labs, dilution 1:600) were used for secondary staining and 4′,6-diamidino-2-phenylindole, dihydrochloride for counterstaining. Images were captured used a digital slide scanner, NanoZoomer 2.0RS (Hamamatsu).

**Statistics.** Results are expressed as mean ± s.e.m. We used GraphPad Prism 4.0c for Macintosh (GraphPad Software Inc.) for statistical analysis. The data was tested for normality using the D'Agostino-Pearson normality test and for equal variance. Differences between two groups were evaluated by Student's $t$-test. If normal distribution or equal variance assumptions were not valid, we used the two-sided Mann–Whitney test. For multiple comparisons, analysis of variance Kruskal–Wallis tests were performed. Animal group sizes were empirically chosen based on prior PET imaging data with nanoparticles. No statistical methods were used to predetermine sample size and animals were not randomly assigned to treatment groups. Data analysis was unblinded. A $P$-value of $<0.05$ indicated statistical significance.

**Data availability.** The authors declare that the data supporting the findings of this study are available within the paper and its Supplementary Information files or can be obtained from the authors upon reasonable request.

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

## Acknowledgements

This work was funded by grants from the National Institute of Health HL096576, HL118440, HL114477, HL117829, HL122208, HL125703, EB009638, K25 EB016673, Deutsche Herzstiftung (S/05/12), American Heart Association (14FTF20380185), NWO Vidi 91713324 and the MGH Research Scholar Award. We thank the Radiochemistry and Molecular Imaging Probes Core at Memorial Sloan Kettering Cancer Center for support (P30 CA008748).

## Author contributions

E.J.K. and Y.-X.Y. performed experiments, analysed data and wrote parts of the manuscript. G.R.W., A.D.A., B.T., M.L.S., H.G., F.F., C.P.-M., C.C., G.C., Y.S., G.C., Y.I., C.W. and H.-Y.W. performed experiments, imaging and analysed data. T.R., J.W.C., F.K.S., J.H., Z.A.F., W.J.M.M. supervised experiments, discussed strategy and data and revised the manuscript. R.W. and M.N. developed the concept, strategy and funding and wrote the manuscript.

## Additional information

**Competing financial interests:** R.W., N.K. and M.N. are inventors on a related patent application. The remaining authors declare no competing financial interests.

**Publisher's note**: 

