## [Peer Review File · Nature Communications]

Reviewers' comments:

Reviewer #1 (Remarks to the Author):

Nahrendorff et al evaluated PET imaging with ¹⁸F-Macroflor, a modified polyglucose nanoparticle with high avidity for macrophages. The authors performed observed that the compound has short blood half-life, measured in three species. They observed accumulation of Macroflor within atherosclerotic plaque macrophages, as well as in inflamed myocardium (post-infarction). Further, the authors performed dual-reported experiments utilizing PET/MRI to simultaneously measure MPO-Gd (an MR agent) along with Macroflor. Therein, they report substantial differences in tracer activity over time and posit that those differences stem from key differences in the macrophage species targeted by the two tracers, differences that could be utilized to non-invasively monitor macrophage biology.

In general, this is a well-performed study that provides important insights. The authors employed robust methodology to conduct the study. The use of multiple animal models enhanced robustness of the findings. The writing is outstanding.

There is no doubt that imaging inflammation could prove to be critically important for future clinical and translational research, and may also prove to be useful clinically. Thus far, FDG is most often utilized for that purpose, though its relative lack of specificity is well described. Accordingly, there is a great need for new agents that are both sensitive yet more specific than FDG. The data provided herein suggest that Macroflur may represent such an agent.

Several clarifications would enhance interpretation of the data.

1) the authors should provide data that may shed light on mechanism by which Macroflor accumulates in macrophages.

2) The authors seem to suggest that macroflor may be subject to less myocardial uptake than FDG when they state: "The difference in myocardial uptake between Macroflor and ¹⁸F-FDG, 167 shown in Fig. 4H, may have implications for cardiac PET inflammation imaging." This is indeed intriguing and would be an important potential advantage of macroflor compared to FDG. However, the authors show a single example image without describing the group differences in myocardial tracer uptake. Could the authors provide the group data?

3) Another intriguing observation made by the authors is the distinctly different temporal patterns of accumulation after my for macroflor vs MPO-Gd. Again, data demonstrating differences in the characteristics of macrophages that accumulate macroflor vs MPO-Gd are needed in order to put the findings into clearer context.

4) The graphs and figures are very well arranged, as is often the case with this group of authors. My minor suggestion is to label the structures in figure 3B

Reviewer #2 (Remarks to the Author):

Kaliher et al developed a new polyglucose nanoparticle, F-18 labeled Macroflor, which exhibits a fast blood clearance and renal excretion. Macroflor enables in vivo PET and optical imaging and quantification of inflammation associated macrophages. The authors performed a concise study using three different animal models/species and correlated their in vivo imaging results to ex vivo experiments. Keliher and colleagues show that primarily macrophages take up the nanoparticle, compared to e.g. neutrophils. The authors used PET and PET-MR for their imaging studies, quantified the in vivo nanoparticle uptake and compared it to FDG uptake. Overall the studies were nicely performed and well presented. The results are of interest and the topic is without any doubts important.

The novelty of this work is limited as it has been shown many times before that macrophages – as it is part of the phagocytes job – take up nanoparticles. However, the new part of this work is that the nanoparticles were reduced in size (5 nm compared to USPIOS which are 15-30 nm; studies 10+ years back). As a logical consequence they exhibit a short blood half-life and fast renal clearance. This leaves also room for further questions as a fast blood clearance gives also less chance for the uptake.

There should be a trade-off between nanoparticle size, blood clearance/background signal and efficient uptake; specifically if targets outside the vasculature (myocardium) are also of importance. The authors perform no comparative studies using different nanoparticles. There is no discussion about limitations of tissue penetration which shall even for 5 nm small nanoparticles dominated by the EPR effect. Thus, a critical appraisal would be appropriate that tissue penetration might be still limited – discuss extremely low uptake in myocardium fig 5E.

Especially in atherosclerotic plaque imaging F-18-Fluoride imaging has been discussed. Thus, the authors should include information about the stability of the nanoparticle radiolabeling in vivo to exclude F-18 instead of nanoparticle uptake.

In respect of novelty the authors also have missed to discuss their nanoparticle approach (unspecific uptake by phagocytes) with recent work performed e.g. by Vogl et al (Nature Communication, 2014) using alarmin/S100A8/9 as specific probe for activated macrophages. Therefore, specifically the discussion section leaves a lot of room for improvement.

The statement in the discussion “FDG target spectrum includes all cells with high glucose uptake, which likely comprises cell types with low Macroflor uptake” would need further validation (e.g. in vitro uptake studies, staining, etc.) as it is clearly speculative.

Minor note:

- Fig 3G has not been references in the result section of the main manuscript

Reviewer #3 (Remarks to the Author):

This interesting paper addresses the development of imaging tools for the identification of “vulnerable” atherosclerotic plaques. The target of this new imaging approach is the identification of macrophage accumulation within plaques by non-invasive means. Nanoparticles, which undergo phagocytosis by macrophages are being labeled with fluor-18. The very small nanoparticles are excreted quickly by the kidney resulting in fast blood clearance. First data suggest that these nanoparticles can be visualized in the atherosclerotic plaque as well as in myocardium which has been ischemically injured. The presented results suggest that Macroflor is a suitable imaging agent for PET/MRI as research and clinical tool to non-invasively monitor macrophage biology.

The authors address a very timely issue. There are a number of radiopharmaceutical approaches proposed to label the macrophages in vivo. The most established approach is the use of F-18-deoxyglucose. However, this approach is hampered by the fact that glucose is utilized by many tissue types. There are nano bodies as well as mannose receptor agents, which are also known to identify macrophage accumulation in tissue. This newly proposed approach adds an additional strategy exploiting the phagocytic potential of macrophages with high affinity.

The use of Makroflor is carefully validated in various in vitro and in vivo models. The overall methodology represents state-of-the art approaches. However, the mouse model is a very challenging model for vascular maging because of its small size. Therefore, the quality of imaging studies are quite limited, especially in the context of thoracotomy. Therefore, it is very difficult to

judge in figure 5, if the “myocardial” activity actually represents inflammation in myocardium or inflammation reflecting wound healing after thoracotomy. However, autoradiography shown also indicates accumulation of the radiopharmaceutical within the myocardium. However, if this activity level is intense enough for imaging, remains questionable. Are there any quantitative results available to convert the signal in % of injected dose.

The data indicate that the radiopharmaceutical is specific for macrophages which appears to be important for possible translation into the clinical environment. How robust and valid this approach is, cannot be seen before a first publication. Also the direct comparison with other imaging approaches such as FDG cannot be performed in this experimental setting. However, as shown in figure 4, there seems to be a close correlation between the regional accumulation of FDG and Macroflor.

The references give appropriate credit to previous work. The methods and results are described in detail, and the abstract summarizes appropriately the data presented. Do all types of macrophages phagocytize these nanoparticles? Are there any toxicological concerns associated with the in-vivo utilization of these particles? The conclusion that this tracer may allow monitoring of macrophage biology may be somewhat exaggerated. The new method appears to track macrophage accumulation in the context of atherosclerotic plaques and inflammation.

Reviewer 1

Nahrendorf et al evaluated PET imaging with ¹⁸F-Macroflor, a modified polyglucose nanoparticle with high avidity for macrophages. The authors observed that the compound has short blood half-life, measured in three species. They observed accumulation of Macroflor within atherosclerotic plaque macrophages, as well as in inflamed myocardium (post-infarction). Further, the authors performed dual-reported experiments utilizing PET/MRI to simultaneously measure MPO-Gd (an MR agent) along with Macroflor. Therein, they report substantial differences in tracer activity over time and posit that those differences stem from key differences in the macrophage species targeted by the two tracers, differences that could be utilized to non-invasively monitor macrophage biology.

In general, this is a well-performed study that provides important insights. The authors employed robust methodology to conduct the study. The use of multiple animal models enhanced robustness of the findings. The writing is outstanding.

There is no doubt that imaging inflammation could prove to be critically important for future clinical and translational research, and may also prove to be useful clinically. Thus far, FDG is most often utilized for that purpose, though its relative lack of specificity is well described.

Accordingly, there is a great need for new agents that are both sensitive yet more specific than FDG. The data provided herein suggest that Macroflor may represent such an agent.

We thank the reviewer for his positive assessment of our work.

Several clarifications would enhance interpretation of the data.

1) the authors should provide data that may shed light on mechanism by which Macroflor accumulates in macrophages.

It is frequently assumed that nanomaterials enter phagocytes, including macrophages, via biologically active processes such as phagocytosis and macropinocytosis (please see Nat Mater. 2014;13(2):125-38). To address the reviewer’s question,

Rebuttal Figure 1 (new Supplementary Fig. 2). Flow cytometry of splenocytes in vitro incubated with Macrolite at 4°C and 37°C. (A) Flow cytometric gating on splenocytes. (B) Histograms of Macrolite fluorescent intensity in macrophages after incubation at 4°C and 37°C. (C) Bar graphs show percentages of Macrolite positive macrophages incubated for 15 minutes and 12 hours at the two temperatures.

we performed an experiment that directly tested the hypothesis that Macrolite enters macrophages via an active cellular process. To this end, we studied in vitro uptake into primary cells harvested from mouse spleens. These uptake studies were done at different temperatures. Splenocytes were incubated with the fluorescently labeled nanoparticle Macrolite for 15 minutes and 12 hours at either 4 or 37°C. Uptake of Macrolite was assessed by flow cytometry, which also identified macrophages by their typical surface marker profile. The experiment was done in duplicates. We found substantial Macrolite uptake only in the cells that were incubated at 37°C, indicating that nanoparticle uptake is inhibited if the cells are on ice (Rebuttal Figure 1). We conclude that Macrolite uptake is a biologically active process. These data were added as new Supplementary Fig. 2.

2) The authors seem to suggest that macroflor may be subject to less myocardial uptake than FDG when they state: “The difference in myocardial uptake between Macroflor and ^{18}F -FDG, 167 shown in Fig. 4H, may have implications for cardiac PET inflammation imaging.” This is indeed intriguing and would be an important potential advantage of macroflor compared to FDG. However, the authors show a single example image without describing the group differences in myocardial tracer uptake. Could the authors provide the group data?

Rebuttal Figure 2 (new Supplementary Fig. 3). Scintillation counting of healthy myocardium in wild type mice 3 hours after injection of ^{18}F -FDG versus Macroflor (Mean \pm SEM, N=7-8, * p <0.01, two-tailed t test).

To address this comment, we compared the percent injected dose per gram myocardium 3 hours after injection of ^{18}F -FDG and Macroflor by scintillation counting. This experiment was done in wild type mice. We found that the uptake of Macroflor is significantly lower than ^{18}F -FDG (Rebuttal Figure 2). These data were added to the manuscript as new Supplementary Fig. 3.

3) Another intriguing observation made by the authors is the distinctly different temporal patterns of accumulation after my for macroflor vs MPO-Gd. Again, data demonstrating differences in the characteristics of macrophages that accumulate macroflor vs MPO-Gd are needed in order to put the findings into clearer context.

We have carefully characterized myeloperoxidase content in myeloid cells previously. The enzyme is more active in neutrophils and the inflammatory monocyte subset (Ly6C^{high} in the mouse). These data were published in *J Clin Invest.* 2010;120(7):2627-34. In the setting of myocardial infarction, we validated MPO-Gd imaging in a serial fashion, and found that the signal intensity reflects the biphasic myeloid response to myocardial ischemia (*Circulation.* 2008;117(9):1153-60). We now revised the discussion as follows to clarify (page 6):

“We previously validated the gadolinium (Gd)-based, activatable enzyme reporter MPO-Gd for infarct imaging⁸ and found the signal to be specific for inflammatory myeloid cells, especially neutrophils and the inflammatory L6C^{high} monocyte subset⁹.”

4) The graphs and figures are very well arranged, as is often the case with this group of authors. My minor suggestion is to label the structures in figure 3B

Thank you -- we added arrows pointing at the increased PET signal in the vasculature.

Reviewer 2

Keliher et al developed a new polyglucose nanoparticle, F-18 labeled Macroflor, which exhibits a fast blood clearance and renal excretion. Macroflor enables in vivo PET and optical imaging and quantification of inflammation associated macrophages. The authors performed a concise study using three different animal models/species and correlated their in vivo imaging results to ex vivo experiments. Keliher and colleagues show that primarily macrophages take up the nanoparticle, compared to e.g. neutrophils. The authors used PET and PET-MR for their imaging studies, quantified the in vivo nanoparticle uptake and compared it to FDG uptake. Overall the studies were nicely performed and well presented. The results are of interest and the topic is without any doubts important.

We thank the reviewer for the positive assessment of our work.

The novelty of this work is limited as it has been shown many times before that macrophages – as it is part of the phagocytes job – take up nanoparticles. However, the new part of this work is that the nanoparticles were reduced in size (5 nm compared to USPIOs which are 15-30 nm; studies 10+ years back). As a logical consequence they exhibit a short blood half-life and fast renal clearance. This leaves also room for further questions as a fast blood clearance gives also less chance for the uptake.

There should be a trade-off between nanoparticle size, blood clearance/background signal and efficient uptake; specifically if targets outside the vasculature (myocardium) are also of importance. The authors perform no comparative studies using different nanoparticles. There is no discussion about limitations of tissue penetration which shall even for 5 nm small nanoparticles dominated by the EPR effect. Thus, a critical appraisal would be appropriate that tissue penetration might be still limited – discuss extremely low uptake in myocardium fig 5E.

Beyond the data presented here, larger dextran nanoparticles have been used for cancer and cardiovascular imaging (e.g. Proc Natl Acad Sci U S A. 2010;107(17):7910-5; Circ Res. 2013 Mar 1;112(5):755-61). Of particular interest is the question whether their accumulation is dominated by the EPR effect. Prior work shows time-dependent macrophage uptake into tumor environments of polyglucose materials (Nat Commun. 2015 Oct 27;6:8692). These effects in perivascular macrophages occur within minutes, and steadily increase over several hours. At a macroscopic scale, tumor uptake of polyglucose nanoparticle derivatives can be substantial, highlighting tumors against background. The low uptake into healthy myocardium is actually an advantage, as it could facilitate imaging of atherosclerotic plaques in coronaries and inflammatory foci such as acutely ischemic myocardium. We have expanded the discussion to refer to the points raised by the reviewer (page 7):

“We therefore correlated ¹⁸F-FDG and Macroflor uptake in rabbits with atherosclerosis. The data show imaging agent overlap. In rabbits with

atherosclerosis¹² and in human endarterectomy specimens¹⁶, ¹⁸F-FDG associates with macrophages detected by histology, which supports the correlation we observed between Macroflor and ¹⁸F-FDG. However, the 0.518 correlation coefficient suggests that both agents behave distinctly. For example, if not sufficiently suppressed with appropriate measures, ¹⁸F-FDG enriches in cardiomyocytes, especially if these are distressed²². This consideration may be relevant for imaging the myocardium and coronary arteries. We observed negligible myocyte uptake of Macroflor, perhaps dictated by the different distribution and tissue penetration of small molecule and nanoparticle based PET imaging agents.”

Especially in atherosclerotic plaque imaging F-18-Fluoride imaging has been discussed. Thus, the authors should include information about the stability of the nanoparticle radiolabeling in vivo to exclude F-18 instead of nanoparticle uptake.

To address this comment, we studied the stability of Macroflor in mouse serum. This experimental setup was chosen because the short blood half life of Macroflor makes in vivo stability studies, especially at relevant later time points, difficult. We therefore opted to investigate Macroflor stability in mouse serum for up to 2 hours. These data were added to the manuscript as new Supplementary Fig. 1.

Stability of Macroflor was examined as described previously (ACS Nano, 2015, 9(4), 3641-3563). Briefly, Macroflor was incubated in mouse serum at 37 °C, and aliquots at different time points were analyzed by size-exclusion chromatography (SEC). Specifically, in 4 x 0.5-mL centrifuge tubes, Macroflor (5 µL, 27.3 ± 0.3 µCi) in 1xPBS was added to mouse serum (20 µL). One sample was immediately injected for SEC analysis while the other three tubes were capped and agitated at 37 °C. At 30, 60 and 120 min each sample was analyzed by SEC. The resulting chromatograms were normalized for relative intensity. The radio-chromatogram indicates that Macroflor is stable for 120 min with no appearance of additional peaks as shown in the new supplemental Fig. 1 (Rebuttal Fig. 3). The single persistent peak at 15.2 min demonstrates that there is no cleavage or degradation of Macroflor.

Rebuttal Figure 3 (new Supplementary Figure 1). Stability of Macroflor in mouse serum. Macroflor was incubated in mouse serum at 37 °C for 120 min, and samples at each time point were injected to radio-SEC for analysis. Data are normalized to relative intensity.

Finally, percent injected dose per gram tissue values obtained in bone by scintillation counting were low (Fig. 1E, F), indicating that minimal to no ^{18}F defluorination of Macroflor occurred.

In respect of novelty the authors also have missed to discuss their nanoparticle approach (unspecific uptake by phagocytes) with recent work performed e.g. by Vogl et al (Nature Communication, 2014) using alarmin/S100A8/9 as specific probe for activated macrophages. Therefore, specifically the discussion section leaves a lot of room for improvement.

We expanded the discussion as suggested and now cite Vogl et al.

The statement in the discussion “FDG target spectrum includes all cells with high glucose uptake, which likely comprises cell types with low Macroflor uptake” would need further validation (e.g. in vitro uptake studies, staining, etc.) as it is clearly speculative.

We deleted this sentence.

Minor note:

- Fig 3G has not been references in the result section of the main manuscript

We corrected this oversight. The panel is now cited in the result section.

Reviewer #3

This interesting paper addresses the development of imaging tools for the identification of “vulnerable” atherosclerotic plaques. The target of this new imaging approach is the identification of macrophage accumulation within plaques by non-invasive means. Nanoparticles, which undergo phagocytosis by macrophages are being labeled with fluor-18. The very small nanoparticles are excreted quickly by the kidney resulting in fast blood clearance. First data suggest that these nanoparticles can be visualized in the atherosclerotic plaque as well as in myocardium which has been ischemically injured. The presented results suggest that Macroflor is a suitable imaging agent for PET/MRI as research and clinical tool to non-invasively monitor macrophage biology. The authors address a very timely issue. There are a number of radiopharmaceutical approaches proposed to label the macrophages in vivo. The most established approach is the use of F-18-deoxyglucose. However, this approach is hampered by the fact that glucose is utilized by many tissue types. There are nano bodies as well as mannose receptor agents, which are also known to identify macrophage accumulation in tissue. This newly proposed approach adds an additional strategy exploiting the phagocytic potential of macrophages with high affinity.

The use of Makroflor is carefully validated in various in vitro and in vivo models. The overall methodology represents state-of-the art approaches. However, the mouse model is a very challenging model for vascular imaging because of its small size. Therefore, the quality of imaging studies are quite limited, especially in the context of thoracotomy. Therefore, it is very difficult to judge in figure 5, if the “myocardial” activity actually represents inflammation in myocardium or inflammation reflecting wound healing after thoracotomy. However, autoradiography shown also indicates accumulation of the radiopharmaceutical within the myocardium. However, if this activity level is intense enough for imaging, remains questionable. Are there any quantitative results available to convert the signal in % of injected dose.

Thank you for the positive comments on our work. The reviewer correctly points out that the limited spatial resolution of PET and the proximity of the thorax wall which needs to be opened to induce MI in mice can make it difficult to discern whether the increase in the PET signal derives from the infarct or from the thoracotomy. We believe that the in vivo images shown in Figs. 5 and 6 exhibit signal in the infarct scar rather than the thorax wall. In addition, ex vivo scintillation counting, autoradiography, intravital microscopy and flow cytometry illustrate Macroflor and Macrolite accumulation in the ischemic myocardium. Nevertheless, the reviewer makes a valid point. The ultimate proof will be obtained in future large animal experiments, which are currently in planning stages. We added the following sentence to the discussion to address the reviewer’s concern (page 7):

"Large animal infarct imaging is required next, since PET imaging in mouse thoracotomy infarct model reaches limitations in terms of spatially distinguishing inflammation in the body wall from myocardial injury."

The data indicate that the radiopharmaceutical is specific for macrophages which appears to be important for possible translation into the clinical environment. How robust and valid this approach is, cannot be seen before a first publication. Also the direct comparison with other imaging approaches such as FDG cannot be performed in this experimental setting. However, as shown in figure 4, there seems to be a close correlation between the regional accumulation of FDG and Macroflor. The references give appropriate credit to previous work. The methods and results are described in detail, and the abstract summarizes appropriately the data presented.

Do all types of macrophages phagocytize these nanoparticles?

Our serial infarct imaging data in mice with MI indicate that M1 as well as M2 macrophages take up the imaging probe, as the imaging was timed to occur during the inflammatory phase (Day 1-3 is dominated by M1 macrophages) and during the resolution phase (Day 4-7 is dominated by M2 macrophages). We detected high Macroflor uptake at both time points. The text was clarified to highlight this important point (page 6):

“The first imaging occurred on day 2 after coronary ligation, which coincides with the inflammatory phase characterized by an abundance of inflammatory neutrophils, monocytes and macrophage subsets (Fig. 6A). The second imaging session was on day 6 (Fig. 6B), which coincides with the resolution phase of infarct healing, when reparative macrophage phenotypes support tissue healing via crosstalk with fibroblasts and endothelial cells in sprouting neo-vessels. The PET standard uptake value for reporting macrophage numbers increased between day 2 and day 6 (Fig. 6C), a change that indicates an expanding macrophage pool in the healing infarct. The increase in Macroflor PET signal correlates well with previous flow cytometric studies documenting rising macrophage numbers between days 2 and 6 after MI^{7, 10}. The data also imply that macrophages of different phenotypes readily incorporate Macroflor.”

Are there any toxicological concerns associated with the in-vivo utilization of these particles? The conclusion that this tracer may allow monitoring of macrophage biology may be somewhat exaggerated. The new method appears to track macrophage accumulation in the context of atherosclerotic plaques and inflammation.

We are currently in the planning stages for formal toxicology studies needed for a phase 1 clinical study (please see last sentence of the discussion). Preliminary toxicity studies in our own labs have been negative (tissue histology, serum enzyme levels). Furthermore, we are optimistic that formal toxicity studies will show similar results, in part also because the nanoparticle synthesis was designed based on clinically approved building blocks.

REVIEWERS' COMMENTS:

Reviewer #2 (Remarks to the Author):

The author addressed all remaining concerns regarding methods and results. I still think a more comprehensive discussion about alarmin would be beneficial.

Reviewer #3 (Remarks to the Author):

The authors have adequately answered all questions raised by the reviewers. Although the data in the mouse model suggest uptake of F-18 tracer in the myocardium, no quantitative uptake data are provided to compare such findings with that of other markers for inflammation.

As pointed out by the authors future studies in larger animal models are needed to assess the translational potential of the proposed imaging approach.

Reviewer 1

no comments

Reviewer 2

The author addressed all remaining concerns regarding methods and results. I still think a more comprehensive discussion about alarmin would be beneficial.

We expanded the discussion as follows (page 7, line 29):

Different imaging strategies for monitoring macrophages have been proposed, comprising imaging of adhesion molecules, cell surface receptors and secreted factors executing inflammatory functions⁴. These strategies likely differ in their sensitivity and specificity for macrophage presence, phenotype target range and pharmacokinetics. Some approaches may be sensitive for early cell activation, such as imaging of alarmins¹², while others, including PET imaging of Macroflor enrichment, report on macrophages in all inflammatory stages, even on non-inflammatory tissue resident cells.

Reviewer #3

The authors have adequately answered all questions raised by the reviewers. Although the data in the mouse model suggest uptake of F-18 tracer in the myocardium, no quantitative uptake data are provided to compare such findings with that of other markers for inflammation.

As pointed out by the authors future studies in larger animal models are needed to assess the translational potential of the proposed imaging approach.

Thank you for your helpful comments during the review process. Indeed, we plan on pursuing large animal myocardial imaging next, as pointed out in the discussion.